# High Performance Work Systems, Justice, and Engagement: Does Bullying Throw a Spanner in the Works?

**DOI:** 10.3390/ijerph19095583

**Published:** 2022-05-04

**Authors:** Elfi Baillien, Denise Salin, Caroline V. M. Bastiaensen, Guy Notelaers

**Affiliations:** 1Department of Work and Organisation Studies, Katholieke Universiteit Leuven (KU Leuven), 1000 Brussels, Belgium; caroline.bastiaensen@kuleuven.be; 2Department of Management and Organisation, Hanken School of Economics, 00100 Helsinki, Finland; denise.salin@hanken.fi; 3Department of Psychosocial Science, University of Bergen, 5015 Bergen, Norway; guy.notelaers@uib.no

**Keywords:** workplace bullying, mobbing, high performance work practices, affective events, moderated mediation

## Abstract

High performance work systems (HPWS) have typically been shown to positively influence employee attitudes and well-being. Research in the realm of HPWS has, in this respect, established a clear connection between these systems and employee engagement through organizational justice. In this study, we analyzed if being bullied affects this relationship. Using reasoning from Affective Events Theory (AET), we expected that the positive association between HPWS and engagement through perceptions of organizational justice is impaired by experiences of workplace bullying. Moreover, we expected a remaining direct effect between HPWS and engagement, also attenuated by bullying. Our results in a sample of service workers in Finland (*n* = 434) could not support the moderating role of bullying in the indirect effect. Workplace bullying did, however, impair the remaining direct relationship indicating it disrupts the positive effect of HPWS on engagement. In all, whereas HPWS were found to be beneficial for not bullied respondents, it was associated with decreased engagement for the bullied. Our findings further underscore the importance of preventing bullying in our workplaces, as it may significantly alter the outcomes of positively intended HR practices into an undesired result.

## 1. Introduction

While occupational health sciences have been the more traditional arena for developing insight in employee well-being, contemporary research in the field of human resource management (HRM) has been increasingly oriented towards policies and practices beneficial for well-being as well. In this respect, HRM scholars have to a large extent studied high-performance work systems (HPWS): a set of separate yet interconnected HR practices that aim to increase organizational performance by creating skilled, committed, and dedicated employees [1]. HPWS typically include aspects such as flexible work arrangements, valid selection procedures, performance-based reward systems, job-oriented training, and supporting management practices [2,3].

Research has well documented the favorable outcomes of HPWS in terms of the organization’s financial results as well as in terms of employee motivation, positive attitudes, and well-being [4]. A highly crucial factor in these findings pertains to the employee’s work engagement [5]: HPWS are found to be successful—for example, by increasing performance and lowering turnover [6,7]—through creating “a positive, fulfilling, work-related state of mind that is characterized by vigor, dedication, and absorption” [8] (p. 210). Moreover, abundant research ascribed a significant role to organizational justice as a key mediator in the HPWS-engagement relationship (e.g., [9,10]). That is because HPWS signal the organization’s good intentions with its staff [11] and foster a balanced social exchange relationship between employers and employees [12]. Our current study builds on these earlier established insights. It starts from the indirect association between HPWS and work engagement though organizational justice as an important explanatory mechanism and aims to further merge this existing line of research into occupational health and well-being science.

While many studies in the realm of HPWS have dug into the ‘why’ of these systems, far less is known about contextual aspects molding the process through which HPWS increase engagement via organizational justice [13]. In addition, even more so, the rather few studies looking at such potential moderators are predominantly framed by HRM’s disciplinary foci; testing factors such as trust in the organization [9], trust in senior management [14], perceived power distance [15], the employee’s age [16], and task proficiency [17]. With this, research to date has ignored that employees are part of a social reality including social interactions, and we are in the dark as to whether the established benefits from HPWS on work engagement persist when the employee is confronted with negative social behavior at work. Therefore, we want to examine whether and how the positive chain of HPWS-justice-engagement is molded by such negative social interactions in the form of workplace bullying, a clearly established topic in occupational health and well-being research [18].

Notably, studying workplace bullying as an issue in the indirect association between HPWS and engagement through justice is important. First, while recent HRM research has acknowledged the impact of how employees perceive their work environment and of their needs during the life and career span, the factors researched are predominantly depicted from a managerial angle looking at issues of organizational climate, policy, and staff subgroups. Moreover, the majority of HPWS research has operated from a motivational perspective and mainly investigated the role of positive factors, i.e., job resources, when looking at the HR systems’ impact on well-being and, more specifically, engagement. A profound knowledge about when HPWS contribute to well-being, however, requires insight in various types of moderators, including negative ones. Given that employees are not working in a social vacuum, social stressors that also significantly attack the employee’s work engagement call for attention. In occupational health research, workplace bullying has been established as one of the most detrimental social stressors at work [19], causing a plethora of negative consequences in its targets [20]. Given its severe impact on motivation and health [21], the evident question is whether the HPWS-engagement process still holds when employees are confronted with such a severe social stressor like bullying. Such insight advances knowledge about how detrimental occupational health phenomena may impact on the effects of—positively intended—organizational staff policies; putting it more central in the overall scientific and practical debates on when policies do and do not work. It can also enrich HPWS research and may further attest workplace bullying as a situation that calls for attention in organizations through prevention.

Additionally, we have theoretical reasons to assume workplace bullying could be a significant influencing factor. HRM scholars have typically discussed the relationship between HPWS and engagement through organizational justice using Social Exchange Theory [12] and Signaling Theory [11]. We, however, see that this process can also be clarified by Affective Events Theory (AET) [22]; counterbalancing the former theories that have been applied in research before with a more emotion-driven perspective. That is, HPWS fuel perceptions of organizational justice because employees appraise the events stemming from these practices as helpful in matching their work context with their personal work goals; being the essential aim of HPWS [1,23]). These perceptions of justice entail positive emotions and affect bringing along positive consequences in the form of engagement. However, being targeted with workplace bullying, a situation which has also been linked to AET [24], will bring along negative consequences that impair the positive process stemming from HPWS. Such an event causing strong negative emotions may block the expected effect from the positive emotions part of organizational justice.

In conclusion, our current study adds to the literature by introducing experiencing workplace bullying as a moderator that hampers the HPWS-justice-engagement process. With this, we contribute to (a) an improved insight in the interrelatedness of the positive chain stemming from HPWS with a well-known social stressor entailing a negative situation, and (b) a further integration of the occupational health sciences and the HRM research field allowing us to clarify the importance of bullying also in the light of human capital-oriented practices.

### 1.1. HPWS: Benefits for Work Engagement through Organizational Justice

In all, HPWS are designed to enhance organizational output by empowering the employee and many studies have documented their positive outcomes. While, initially, studies have looked at the benefits of the intended (i.e., the policies and practices as developed at the organizational level) and actual (i.e., the practices as enacted and documented by line management) high-performance HRM practices, scholars have more lately focused their attention on how HPWS are perceived by their ultimate recipient, being the employee [25]. This is because, in the end, the employee’s perception of the practices affects how they are thinking, feeling, and behaving, and whether HPWS will lead to the intended outcomes for the staff and the organization [26].

Drawing on the ‘mutual gains’ perspective—the idea that HPWS add to positive employee-related outcomes on top of organizational performance [27]—scholars found a significant impact of HPWS on employee productivity (e.g., [28]), organizational commitment (e.g., [29]), organizational citizenship behavior (e.g., [30]), proactive behavior (e.g., [31]), and decreased turnover (e.g., [32]). Moreover, a plethora of studies indicated that HPWS contribute to increased well-being in the form of work engagement (e.g., [33,34,35]). The employee’s work engagement has been regarded as vital to HPWS as these systems are explicitly designed to have a positive effect on engagement and, in turn, performance [5,36]. In fact, one study detected that, the more well-being-oriented factor of, work engagement offered a more comprehensive explanation of performance as compared to job involvement, job satisfaction, and intrinsic [37].

Abundant research has explored the ‘why’ of HPWS’ impact on positive outcomes, and for work engagement it has pointed at organizational justice as a key explanatory mechanism: HPWS increase the employee’s justice perceptions that, in turn, bring along this perceived work engagement [9]. Organizational justice captures the extent to which employees perceive organizational events as being fair [38] and is typically manifested through three types. Procedural justice—the perceived fairness of decision-making procedures [39]—relates to a transparent decision-making process including the employee’s participation, which is the essence of HPWS. Interactional justice—a fair interpersonal treatment received from the employee’s managers during these procedures—focuses on social sensitivity and informational justification [40]. Such communication, being it to clarify the arguments behind a decision or to signal that management is receptive towards the employee’s input, is also part of HPWS. Finally, distributive justice—the perceived fairness of rewards—is high when the employee receives the correct rewards for the work that has been done, as compared with others in the organization [41]. HPWS integrate many performance-based practices balancing the effort—reward relationship. From a more general perspective, scholars have explained the HPWS-justice-engagement relationship using the Social Exchange Theory [12] that postulates an exchange-relationship between employers and employees. When the organization provides substantial inducements to its employees, they are more likely to reciprocate positively in attitudes and well-being [42]. Additionally, HPWS may signal the organization’s good intention with their employees (Signaling Theory) [43], as such contributing to their well-being.

While a great number of studies investigated explanatory mechanisms, the literature on ‘when’ the HPWS foster positive well-being is far less developed. Moreover, this literature focuses on contextual factors fitting HRM scholars’ interest in what strengthens the positive effects of HPWS. For example, building on the proposition that trust molds the association between an interaction partner’s positive action and the receiver’s response, Farndale and colleagues [9] found empirical evidence of the boosting role of the employee’s trust in the organization. Similar results were found for a higher trust in senior management and a lower perceived power distance within the organization [14,15]. A study interested in a possible age-related difference in the positive outcomes of HPWS showed no significant impact of age [16] and concludes towards a seemingly robust desirable effect of HPWS for employees of all ages. Boon and Kalshoven [17] were among the first to indicate that HPWS are especially important for motivating employees who experience challenges in having sufficient skills and abilities in their work (i.e., low task proficiency) as rated by their supervisor. With this, they point at the role of these systems in getting the ‘weaker’ pawns in the organization aligned towards the organization’s goals. While this research has shed some light on how context can influence the HPWS process towards engagement, it falls short in tapping issues that, in the respect of engagement as a well-being outcome, are of particular interest from an occupational health perspective. More specifically, while employees are individuals with certain skills (or not) embedded in an organizational context, they work and interact with others and thus experience a social reality. Drawing on the earlier findings related to, for instance, the boosting impact of perceived trust and power distance in the organization [44], we can rather confidently argue for a beneficial impact in this relationship when looking at indicators of a positive social climate. In contrast, we do not know whether and how exactly the association of HPWS with engagement through justice is influenced when employees are confronted with significant negative events such as workplace bullying. Will the positive chain prevail even under such detrimental circumstances, or will these negative social behaviors block the employee’s opportunities to reap the benefits of HPWS?

### 1.2. Interference by Workplace Bullying

While HPWS have generally been regarded as a positive investment from the organization in its employees and while studies have pointed at the many advantageous consequences of these practices, we thus ask ourselves whether these constructive outcomes remain when employees are confronted with workplace bullying. Workplace bullying refers to interpersonal mistreatment in which an employee is repeatedly targeted with negative social acts at work [45]. While many of these negative acts—including gossiping, spreading rumors, or withholding information—may not be problematic in isolation, they can cause severe harm when an employee experiences these in combination and over a longer period of time (e.g., six months) [46]. Consequently, workplace bullying has been shown to cause impaired well-being, such as physical health problems, burnout, symptoms of post-traumatic stress, increased intentions to leave, absenteeism, reduced job satisfaction and reduced organizational commitment [20]. While bullying can be enacted by any of the organizational members [47], it is typically characterized by a power disparity: the target experiences difficulties in defending him or herself against the perpetrator’s negative social behaviors [48,49].

As to why bullying brings along these negative effects, scholars have argued that it should be considered as an affective event [24,50,51]: experiences of bullying elicit emotions such as fear, anger, irritability, and shame [52,53] that could mold the plethora of negative outcomes in its targets. This reasoning ties in with Affective Events Theory (AET) [22] from which we can derive that what happens at work shapes the employee’s attitudes and well-being through not only cognitive but also emotional information processing. Given that workplace bullying confronts its targets with prolonged negative social acts that threatens the employees’ overall functioning as well as self-esteem and social belongingness [50], a range of studies have successfully applied AET as a framework explaining negative outcomes such as decreased engagement, lowered job satisfaction, higher intention to leave the organization, organizational commitment. AET could even account for notable detrimental outcomes such as accidents and injuries in healthcare (e.g., [24,51,54,55,56]).

Interestingly, however, while HRM scholars have mostly looked at the HPWS-justice-engagement chain from the perspectives of social learning [12] or signaling theory [43], we also see an obvious link with AET. More specifically, AET postulates that work attitudes and responses—such as engagement—stem from an accumulation of affective responses elicited by the work environment [22]. In this process, the employee evaluates work events as being helpful versus harmful to reach their relevant goals. Helpful events bring along positive feelings, whereas events hampering goal process result in negative feelings. Then, the employee considers additional details about the events (e.g., who is responsible, or can it be easily addressed) that leads to more specific emotions (e.g., joy, fear, or anger) [22]. Applying this to HPWS, justice, and engagement, it is clear that HPWS are designed with the aim of reaching work goals [23] that—according to AET—add to positive emotions manifested in perceptions of organizational justice [9]. This is because, whereas injustice relates to negative emotions and to undesirable outcomes, justice entails positive emotions and positive outcomes [57,58]. In other words, HPWS are helpful for the employees in reaching their goals and manifest themselves through the positive affective state of perceived organizational justice and the positive outcome of work engagement. Taken together, the association between HPWS and work engagement through organizational justice ties in with a positive affect process, while workplace bullying entails a negative affect process that, as a social, relational stressor, may attenuate the HPWS-justice-engagement chain. From this, we formulate following hypothesis:

**Hypothesis** **1:***The indirect association between HPWS and engagement through organizational justice is buffered by the experience of workplace bullying (i.e., moderated mediation)*.

Notably, organizational justice is just one possible manifestation of positive affect and other factors could also be at stake as a potential explanatory mechanism in the link between HPWS and engagement. Consequently, in addition to the indirect association through justice, we still expect a direct relationship between HPWS, and engagement remains which, from an AET lens, could still be impacted by the negative event of workplace bullying; and also include this in our analyses. We, thus, assume:

**Hypothesis** **2:***The remaining direct association between HPWS and engagement is buffered by the experience of workplace bullying (i.e., moderation)*.

Our research model is depicted in Figure 1.

## 2. Method

### 2.1. Procedure and Sample

A survey was conducted in Finland among service workers (*n* = 434) in collaboration with Service Union United PAM; a Finnish trade union for people working in private service sectors. PAM has almost 200,000 members in total, and these are employed across a large number of private organizations, in sectors such as retail trade (largest sector), hotel and restaurant services, cleaning and property services, and security services. The sample size was determined in negotiation with the union. The research director of PAM distributed an online version of the survey to 5000 randomly selected members. The recipients received a cover letter and a link to the survey. As Finland is a bilingual country, the survey was available in both Finnish and Swedish and the respondents themselves could choose the language.

The sample’s mean age was 39 years (SD = 11.69), ranging from 17 to 63 years. About 78% of the participants were female, and 10% of the participants held a supervisory position. Regarding tenure, 34% of the participants were employed within their current organization for more than 10 years, and 20% had worked for their employer for less than one year.

### 2.2. Measures

All concepts part of our research model were measured using internationally validated scales.

High-performance work systems (HPWS) (α = 0.93) were assessed using 24 items from Chuang and Liao [59]. On a five-point Likert scale ranging from ‘strongly disagree’ (=1) to ‘strongly agree’ (=5), the respondents replied on statements related to six different areas of HR: staffing (e.g., “Recruitment emphasizes traits and abilities required for performing well in this organization”), training (e.g., “My organization invests considerable time and money in training”), performance appraisal (e.g., “Performance appraisals are based on objective, quantifiable results”), compensation (“Employee salaries and rewards are determined by their performance”), participation (e.g., “If a decision made might affect employees, the organization asks them for opinions in advance”), and caring (e.g., “My organization has formal grievance procedures to take care of employee complaints and appeals”).

Organizational justice (α = 0.88) was measured using eight items from Elovainio and colleagues [60]. The respondents indicated on a five-point Likert scale to what extent they agreed (‘strongly disagree’ = 1; ‘strongly agree’ = 5) to statements tapping procedural justice (e.g., “I can express my views and feelings when decisions are made/procedures are applied”), interactional justice (e.g., “My supervisor tailors his/her communications to individuals’ specific needs”), and distributive justice (e.g., “My compensation reflects the effort I have put into my work”).

Five items from Utrecht Work Engagement Scale (UWES) [61] were used to measure work engagement (α = 0.94). The items were addressed using a seven-point Likert scale ranging from ‘never’ (=0) to ‘a few times a year or less’ (=1), ‘once a month or less’ (=2), ‘a few times a month’ (=3), ‘once a week’ (=4), ‘a few times a week’ (=5) and ‘every day’ (=6). Example items are “At work, I feel bursting with energy” (vigor), “When I get up in the morning, I feel like going to work” (dedication) and “I am immersed in my work” (absorption).

The experience of workplace bullying behaviors (α = 0.93) was assessed using the Short Negative Acts Questionnaire (S-NAQ) [62]. Respondents had to indicate how often, during the last 6 months, they experienced nine bullying behaviors (e.g., “Silence or hostility as a response to your questions or attempts at conversations”). These items were tapped using a five-point Likert scale ranging from ‘never’ (=1) to ‘now and then’ (=2), ‘monthly’ (=3), ‘weekly’ (=4), and ‘daily’ (=5).

### 2.3. Plan of Analysis

Overall, most scholars have applied (mean) sum scores, standard deviations, and analyses of variance when studying workplace bullying. However, bullying typically follows a negative binomial distribution. While valuable in shedding some light on this phenomenon, these more dominantly used techniques generally assume a normal distribution and could therefore challenge their statistical conclusion validity in terms of workplace bullying [63]. Therefore, in this study, we modeled experiencing workplace bullying as several exposure categories (i.e., categorical variable). We followed the upcoming statistical approach in bullying research by applying a Latent Class Cluster Analysis (LCCA) technique: studies using LCCA detected qualitatively different clusters (subgroups of respondents) each showing a different combination and frequency of the various negative social behaviors measured [62,64,65]. Notably, LCCA has particular advantages as compared to classical clustering techniques (e.g., K-means): as a model-based approach it allows for statistical tests in determining the number of clusters [66], and it is insensitive for different variances in the items part of the measurement [67] which suits the S-NAQ [62]. Therefore, in our analyses, we first identified different profiles of bullying using LCCA in Latent Gold 5.0 (Statistical Innovations, Arlington, TX, USA). Then, we tested our hypotheses using Hayes’ [68] PROCESS macro v3.5 (model 15) in SPSS 25 (IBM, Armonk, NY, USA) in which we introduced workplace bullying as a categorical variable accommodating for the possible presence of different profiles.

## 3. Results

### 3.1. Identifying the Targets of Bullying

LCCA first groups all respondents into one cluster, and sequentially adds clusters until a measurement model is found that fits the data best [67]. Model fit is assessed based on the Bayesian Information Criterion (BIC; this should be low), L2 (using bootstrapping following Langeheine, Pannekoek, & Van de Pol [69]; this should be non-significant), the total amount of bivariate residuals (BVR; should be low), and the bivariate association between the indicators (reduced with at least 85%). Finally, the reduction in L2 signals how much of the association between the indicators is explained by adding an additional cluster.

Table 1 lists the statistics of the LCCA for our data, supporting us in selecting the best clustering for our data. First, the criteria indicated that our respondents should be allocated to a number of latent class clusters (instead of one). However, a close inspection of the profiles showed that, from the point that four clusters had been identified, Latent Gold kept extracting extra clusters between the response categories of 1 (never) and 2 (occasionally) without much extra change in the fit criteria, which led us to focus on the first 5 cluster solutions. In the 5-cluster solution, the total amount of bivariate residuals (BVR) decreased strongly, from 7111 to 39.2. However, already in the 4-cluster solution, 99% of the residuals had been accounted for. In addition, the total amount of BVR in the 4-cluster solution was significantly lower (ΔBVR = 61.3) than in the 1-cluster solution. A detailed look at the bivariate associations between the indicators showed that, compared to the 1-cluster model, all bivariate residuals were reduced with a least 97%. Finally, the bootstrap of the L2 was not significant (*p* = 0.124); and the decrease in L2 notably declined from the 3-cluster to the 4-cluster solution (ΔL2 = 152.7) with ΔL2 from the 4-cluster to the 5-cluster solution reaching only 113. Combining these points—the extra extraction of theoretical less relevant latent clusters when extracting 5 or more clusters combined with satisfactory fit criteria—we concluded that four clusters are sufficient and best suitable to our data.

The first cluster (41%) entailed the ‘not bullied’ with showing an average conditional probability of approximately 0.85 to respond ‘never’ to the NAQ-items. The second cluster (40%) were ‘rarely confronted with negative encounters’: their average conditional probability to respond ‘never’ to the items was still 0.25; however, that of responding ‘occasionally’ was approximately 0.50. Their average probability to respond ‘weekly’ or ‘daily’ was less than 0.05. In the third cluster, the ‘occasionally bullied’ (16%), the average probability to respond ‘never’ to the items was close to 0.05. Yet, the average probability of responding ‘occasionally’ or ‘monthly’ was approximately 0.55. The fourth cluster (2%) consisted of ‘severe targets’ of bullying. In this group, the conditional probability to respond ‘never’, ‘occasionally’, or ‘monthly’ to the items is nearly zero. The conditional probability of responding ‘weekly’ or ‘daily’ exposure to the negative acts is close to or higher than 0.90. In all, the analysis showed that there are four different latent profiles reflecting a certain exposure level to bullying. These will be used as the moderator in the subsequent analysis.

### 3.2. Test of Hypotheses

Table 2 summarizes the means, standard deviations, and correlations of our measurements. Overall, HPWS correlated positively with organizational justice and work engagement. Justice correlated positively with engagement. Notably, the probability of being ‘not bullied’ associated positively with HPWS, justice, and engagement. The probability of being ‘rarely confronted with negative encounters’ correlated negatively with HPWS and justice but was unrelated to engagement. The probability of being ‘occasionally bullied’ was negatively related to HPWS, justice, and engagement. Being a ‘severe target’ correlated negatively with justice and engagement, and not with HPWS. These results give a first, more nuanced view on bullying in the context of our study, pointing at the importance of approaching this phenomenon as different exposure groups.

We tested our research hypotheses introducing experiencing workplace bullying as a categorical variable in line with the LCCA results. More specifically, bullying was included in the analyses using the following reference coding: (1) ‘rarely confronted with negative encounters’ as compared to the other latent clusters, (2) ‘occasionally bullied’ as compared to all other clusters, and (3) severe targets as compared to the other clusters. The model (see Table 3) explained 32.67% of the variance in engagement: 29% was accounted for by the main effects, while the moderation of the direct paths between HPWS and engagement accounted for 4.69% of the variance explained. The moderation of the indirect path between HPWS and engagement through organizational justice was not significant, with the absolute value of the Index of Moderated Mediation being smaller than 1.96 times the bootstrapped standard error (boot se) (‘rarely confronted with negative encounters’: −0.014, boot se of 0.192; ‘occasionally bullied’: 0.015, boot se of 0.273; ‘severe target’: 2.506, boot se of 2.951); rejecting hypothesis 1.

However, and interestingly, when looking at the exposure groups specifically, the indirect effect of HPWS and engagement through organizational justice was significant for the ‘not bullied’ (0.463 **), ‘rarely confronted with negative encounters’ (0.448 **), and ‘occasionally bullied’ (0.478 **). For the ‘severe targets’, organizational justice did not mediate the HPWS-engagement relationship. Our results did reveal a significant interaction of bullying on the remaining direct relationship between HPWS and engagement. Specifically, for the ‘not bullied’ the relationship was not significant (b = 0.268; *p* = 0.170). For the ‘rarely confronted with negative encounters’, this relationship was positive and significant (b = 0.876; *p* < 0.001). In contrast, this relationship was slightly negative—yet not significant—for the ‘occasionally bullied’ (b = −0.422; *p* = 0.157). Finally, among the ‘severe targets’ the relationship was strongly negative and significant (b = −2.847; *p* < 0.001). These findings correspond with hypothesis 2, and further nuances it in terms of the bullying exposure groups: whereas HPWS were beneficial for the engagement of the ‘non bullied’ and those ‘rarely confronted with negative encounters’, HPWS related to decreased engagement for the ‘severe targets’.

In all, the results confirmed a positive relationship between HPWS and engagement, and suggested that workplace bullying acts as a moderator (note that including gender (0 = male; 1 = female), tenure (in years), and supervisory position (0 = no; 1 = yes) in our analyses did not alter our results and conclusions). More precisely, when employees are subjected to high levels of bullying, the positive relationship between HPWS and work engagement diminishes.

## 4. Discussion

The current study aimed to shed light on whether and how the social stressor and negative affective event of workplace bullying molds the manifoldly reported positive association between HPWS and work engagement through organizational justice in HRM research. With this, we introduced an important impairing occupational health issue as to further knowledge on positively intended organizational staff policies and practices. Our study advanced the bullying literature by drawing focus to this form of interpersonal mistreatment, as a social stressor, in the overall scientific debates on when HRM policies do and do not work. Moreover, we adhered to notable considerations regarding the statistical conclusion validity of existing research that has approached workplace bullying through sum scores in analyses of variance: we tied in with the evolution of modeling bullying as several exposure categories established through Latent Class Cluster Analysis [63]. Finally, our study also contributed to HPWS research by responding to several calls for more research on individual-level conditions under which HRM affects employee attitudes [13,15].

In all, our results tie in with the established indirect association between HPWS, organizational justice, and engagement (e.g., [9,10]); yet—contrary to our expectations—we could not detect a significant moderation of the workplace bullying exposure categories. From this, we could derive that the positive chain of HPWS-justice-engagement is not impacted by events of workplace bullying. However, digging into the more detailed situation for each of the bullying exposure groups, we can further nuance this. More specifically, for the employees belonging to the ‘not bullied’, ‘rarely confronted with negative encounters’, and ‘occasionally bullied’ groups, the indirect effect was significant and, thus, remains. This is in contrast with employees in the ‘severe target’ group for whom HPWS did not relate to organizational justice and, subsequently, engagement. In other words, for employees frequently experiencing these negative social behaviors at work, the organization’s investment in HR practices to create committed and engaged employees through increased feelings of justice [1] are not paying off. From a bullying perspective, these findings are highly intriguing as scholars have been trying to gain a better understanding in how exactly organizational justice and its consequences can be grasped in the light of bullying. While some identified bullying as a consequence of injustice (e.g., [70,71]), others have looked at justice as a buffer protecting bullying victims from negative well-being (e.g., [72]). In a recent study of 280 cases, Neall, Li, and Tuckey [73] could add another perspective to this debate: from formally reported bullying, they saw that this—by the target described events of low justice in response to their several complaints as part of a bullying case—fueled further escalation of the bullying and its consequences for these targets. From this and looking at our own results, we might consider the idea that employees yield a qualitatively different interpretation of and focus on their work context, depending on which exposure group they belong to, molding our observed effects. That is, employees belonging to the not-bullied categories (the largest group with no or very limited social issues in this respect) may not be inclined to reflect more critically on the organization’s practices, leading them to acknowledge the HPWS as stemming from good intentions and to perceive organizational justice [74]. In this situation, HPWS was related to engagement through justice. The same was so for the occasionally bullied who might not be questioning the overall organization in the light of their situation, yet could be more drawn towards sensemaking in terms of their more direct social interactions (for example, by attributing the situation to one or more ‘bad apples’) [75]. In contrast, severe targets might have been stranded in a situation in which the nature of bullying directs their sensemaking to negative issues sustaining the events (i.e., no significant indirect path). Clearly, there still is much to unravel when it comes to our understanding of justice in the context of workplace bullying.

While the impact of the bullying exposure groups on the indirect association was non-significant, we did find an interaction of bullying on the remaining direct relationship between HPWS and engagement. Again, we see quite an interesting pattern when looking at the groups separately: whereas the relationship between HPWS and engagement was non-significant for the ‘non-bullied’, it was positive for those ‘rarely confronted with negative encounters’, non-significant for the ‘occasionally bullied’, and negative for the ‘severe targets’. These findings support our assumption that bullying can be regarded as a disruptive social factor in reaching the HPWS aims. Interestingly, from our more overall results, it seems that for the ‘non-bullied’ the link between these HR practices and engagement can entirely be explained by justice. Or, when not being confronted with negative social behavior at all, the HPWS do relate to higher perceived justice and, following, engagement. For the ‘rarely confronted with negative encounters’ HPWS related to engagement through justice and directly. The ‘occasionally bullied’ follow the results for the ‘non bullied’, yet they are the first group to show a shift in the remaining direct association between HPWS and engagement. Finally, the severe targets never benefit from the HPWS; neither indirectly nor in the remaining direct effect as these practices will, for them, strongly decrease their engagement. In all, we may conclude from our study that, whereas HPWS are beneficial for the engagement of the non and rarely confronted with negative encounters, they decrease the engagement for the bullied.

In formulating our hypotheses, we built on established knowledge on why HPWS may mold engaged employees and subsequently ‘work’ for the organization in terms of performance and much-desired results. Looking at organizational justice as an important mediator in this respect, we broadened the existing theoretical lenses in the HPWS research stream—Signaling Theory [43] and Social Exchange Theory [12]—with Affective Event Theory (AET) [22]. Using AET as a shared framework in explaining effects of both HPWS and bullying helped us in merging and further contextualizing studies that have been conducted in very separate fields of research. Notably, as also indicated when explicating our research hypotheses, justice is but one possible manifestation of positive affect, and other mediators—now part of the remaining direct effect—might play a role as well. Some examples in this respect could perhaps be perceived organizational support or even rebalancing the employee’s psychological contract from transactional to a more relational one [29,44]. However, while AET largely focuses on the affective process behind human responses to situations, it also yields a less developed cognitive path. Or, while events trigger outcomes through affective states, the theory still acknowledges that they could do so through cognitions. From this angle, an explanation for our current findings regarding the direct effect might be a structural aspect: one conceptual paper proposed that HPWS relate positively to what they merge under the term of internal social structure—including facilitating network ties, generalized norms of reciprocity, shared mental models, and role making/taking—thereby reaching positive employee attitudes, well-being, and performance [13]. Going back to the characteristics of the several bullying exposure groups and our results on the direct moderation, we could reason that the components defined as part of this internal social structure are in fact highly challenged because of the pattern of negative social behaviors experienced in the severe target group. From a more meta-theoretical perspective, we could then also think along the lines of Conservation of Resources Theory (COR) [76]. Central to COR are resources, defined as ’those objects, personal characteristics, conditions, or energies that are valued by the individual or that serve as a means for attainment of these objects, personal characteristics, conditions, or energies’ [77] (p. 516). Overall, resources add to positive outcomes such as growth and well-being. A (threat of) resource loss brings along energy depletion, stress, and negative outcomes. Studies to date have successfully applied COR to bullying and their findings established that experiences of bullying trigger a process of resource loss (e.g., [78,79]). Subsequently, adhering to the idea of an improved internal social structure when installing or promoting HPWS, bullying may well be depleting the employees’ accessibility to the beneficial resources part of such a structure. This might explain why the direct association between HPWS and engagement shifted from a positive to a negative association for the ‘occasionally bullied’.

In all, from our research we may derive some important points. First, investigating which stress-inducing individual-level events or social situations impact on well-intended organizational policies and practices for improving employee engagement and motivation matters. We were the first to sketch a more nuanced image on HPWS as such a practice, combined with bullying as a social stressor. Notably, the significance of combining motivational and stress-related processes in explaining a range of outcomes has been underscored already many decades ago by Karasek [80]. With this study, we want to encourage scholars to further integrate insights from the occupational health and well-being field into better understanding the impact of motivational organizational policies. Second, looking at bullying as a phenomenon entailing qualitatively different exposure clusters is important. Our findings underscore that the reality when it comes to being bullied is far more complex—with differentiated results over the exposure groups—than a story of low versus high exposure. Research on workplace bullying will undoubtedly benefit from applying a LCCA lens for truly fine graining this form of interpersonal mistreatment.

### 4.1. Limitations and Future Research

As with any research, our study entails some limitations that would be valuable to being addressed in future research. First, we have built on cross-sectional data; implying we could not unravel time-based associations between the concepts part of our study. As we build on the much-established chain of HPWS-organizational justice-engagement, we see no grand issues for the indirect effect. Moreover, given that we are the first to introduce an individual-level social stressor in this indirect chain, we tied in with Spector [81] who indicated that “it makes sense to start new areas of inquiry with the most efficient methods to provide initial evidence that a research question is deserving of attention” [81] (p. 129). Our findings have surely contributed to some first insights on the influence of bullying in HPWS and their presumed beneficial effects. Nevertheless, while having lifted a corner of the veil, future studies could progress our current understanding by longitudinally and more dynamically exploring where in time being bullied plays a role in the indirect process, which might even be different for various exposure groups.

Second, we relied on single-source, self-reports and, consequently, our results could have been impacted by common method bias [82]. However, self-reported measures are suitable in this study, given our explicit aim to investigate (a) how HPWS are perceived by the employee (following evolutions in this research stream) [25], (b) how this relates to perceived organizational justice [83], and (c) the employee’s own experience of being bullied. Regarding bullying specifically, meta-analytical evidence [84] underscored that self-reports provide a more reliable and valid assessment of mistreatment than did other-reports when surveys were anonymous, which was the case in our study here. Moreover, we have followed further recommendations to diminish common method bias by, for example, emphasizing the voluntary nature of this study and by ensuring the respondents that there were no correct or wrong answers [82]. In addition, we inspected method bias by comparing the fit of our theoretically expected factor model—χ^2^(939) = 1738.84, *p* <0.001; CFI =0.94, TLI = 0.93, RMSEA = 0.04—to a single factor test (Harman, 1979) which produced a significantly poorer fit to our data; χ^2^(989) = 7733.17, *p* <0.001; CFI = 0.46, TLI = 0.43, RMSEA = 0.13. In addition, the more advanced approach including a common method factor did not reveal an improved statistical fit. This is because, given the large number of latent factors to be estimated in this latter approach on our sample of 434 respondents, the analyses failed to estimate the model’s standard errors and is therefore poorly defined.

Finally, interaction effects are hardly induced by method bias [85] which is, in fact, more likely to attenuate rather than to strengthen interactions [86].

Third, some other limitations relate to our sample. A vast majority of our participants worked in retail trade and that sector was clearly overrepresented compared with other subsectors (when looking at where PAM members in general work). Women made up 78% of the sample compared to 76% in PAM overall so—even while our study’s sample is dominated by female employees—this seems to reflect the real gender distribution quite well. Relatively few (11 out of 434) reported another native language than Finnish or Swedish, suggesting immigrants may have been less likely to respond (5% of all PAM members according to their own reports). In addition, our sample was relatively small in size, due to which we analyzed the mean scores of HPWS and organizational justice. In addition, that hypothesis 1 was rejected while the interaction effect added 1% to the explained variance of engagement might question of whether our study was somewhat underpowered. Future studies could collect larger and more heterogeneous samples that allow for testing the different aspects of HPWS and organizational justice, for further knowledge on how exactly bullying impacts on these components in view of employee well-being.

Finally, our reasoning builds on AET [22] without explicitly testing the several components of this theory. Future research could therefore see how AET could be more explicitly applied when integrating the HRM and occupational health and well-being research streams. These studies could, for example, specifically tap the emotions and affects at stake in boosting versus attenuating the positive impact of HPWS on well-being from the perspective of bullying, which would be particularly interesting in more dynamic and shortitudinal research designs [87].

### 4.2. Implications for Practice

Our findings show that the association between HPWS and work engagement does not always hold: for those severely bullied, we found a negative relationship between HPWS and engagement. This underscores that social relationships are an important part of the equation regarding the effects of such systems. For organizations having already invested in HPWS, it implies that not only investment in the staff’s motivation and engagement matters, yet that they should similarly work on workplace bullying prevention as to ensure the systems will manifest themselves in the desired outcomes. Organizations considering implementing HPWS are encouraged to first assess their situation in terms of bullying to increase the success of this effort. In other words, investing in occupational health outruns merely putting energy in motivational practices: when organizations are aiming at engagement from their staff, they should be aware of how their policies—implemented with good will—are influenced by more individual social stressors, such as workplace bullying. Moreover, our study draws attention to being aware that bullying is not a continuum. In contrast, when categorizing respondents based on the frequency and the nature of the reported negative social behaviors, we see a so much more nuanced picture of what is happening. Either way, we revealed the first evidence of how important being bullied really is in the context of ensuring organizational efforts for motivating staff, such as HPWS, to translate in desired positive outcomes. Or, to put it differently: when it comes to bullying, prevention really is key.

## 5. Conclusions

Our study points at the importance of being bullied as a negative social event experienced by individual workers in disrupting the positive effect of HPWS on engagement (through organizational justice). With this, we demonstrated that bullying does not only lead to negative effects on target attitudes and well-being as shown by previous research, but that it contextualizes the effects of organizational HR practices explicitly designed to advance employee engagement, motivation, and well-being, and are generally associated with positive employee outcomes. Overall, HPWS were beneficial for not bullied respondents, yet decreased engagement in bullied employees. As such, we further attest the value of workplace bullying prevention in organizations, as it may significantly alter the desired results of positively intended HR practices.

## Figures and Tables

**Figure 1 ijerph-19-05583-f001:**
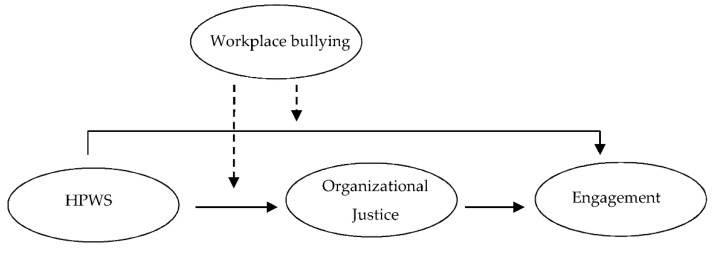
Hypothesized research model.

**Table 1 ijerph-19-05583-t001:** Determining the number of the bullying exposure clusters: LCCA fit statistics.

	BIC (LL)	AIC (LL)	AIC3 (LL)	Npar	L²	Total BVR	VLMR	Class. Err.	Entropy R² in %
1-Cluster	9346.3	9200.5	9236.5	36	4760.4	7111.0	-	0.0	100
2-Cluster	8084.2	7898.0	7944.0	46	3437.9	1050.5	1322.5	2.93	90.19
3-Cluster	7719.5	7492.9	7548.9	56	3012.7	143.6	425.1	5.10	88.17
4-Cluster	7627.3	7360.2	7426.2	66	2860.0	61.3	152.6	5.67	87.26
5-Cluster	7574.6	7267.0	7343.0	76	2746.9	39.2	113.1	8.68	84.14

**Table 2 ijerph-19-05583-t002:** Means, SD, and (auto)correlations of the studied concepts.

	*M*	*SD*	1	2	3	4	5	6
1. HPWP	2.586	0.739	*0.901*					
2. Organizational Justice	3.049	0.890	0.708 **	*0.878*				
3. Engagement	5.030	0.527	0.419 **	0.491 **	*0.923*			
4. Probability to be not bullied	0.416	0.466	0.303 **	0.488 **	0.274 **	-		
5. Probability to be rarely exposed	0.402	0.445	−0.132 **	−0.230 **	−0.071	−0.680 **	-	
6. Probability to be occasionally bullied	0.159	0.343	−0.198 **	−0.292 **	−0.219 **	−0.415 **	−0.312 **	-
7. Probability to be a target of bullying	0.024	0.151	−0.093	−0.164 **	−0.142 **	−0.140 **	−0.141 **	−0.067

Note. **: 0.001 ≤ *p* < 0.01. Autocorrelations are presented in italics.

**Table 3 ijerph-19-05583-t003:** Results of the Moderation Mediation Analyses for engagement, including workplace bullying as a categorical variable (based on LCCA).

Predictors	Unstandardized Beta	R^2^
Intercept	5.157 ***	
HPWS	0.268	
Organizational Justice	0.541 **	
Rarely confronted with negative encounters	−0.043	
Occasionally bullied	−0.591 *	
Severe target	0.013	29.03
HPWS * rarely confronted with negative encounters	0.608 *	
HPWS * occasionally bullied	−0.689 *	
HPWS * severe target	−3.115 ***	4.69 ***
Organizational Justice * rarely confronted with negative encounters	−0.016	
Organizational Justice * occasionally bullied	0.018	
Organizational Justice * severe target	2.933 *	1.05
Total		32.67

Note. (*): 0.05 ≤ *p* < 0.10; *: 0.01 ≤ *p* < 0.05 **: 0.001 ≤ *p* < 0.01 and ***: *p* < 0.001.

## Data Availability

Not applicable.

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
