# Peer review of "High Performance Work Systems, Justice, and Engagement: Does Bullying Throw a Spanner in the Works?"

_ijerph, 2022, doi:10.3390/ijerph19095583_

Round 1

Reviewer 1 Report

It is an interesting and original paper addressing bullying at work as a factor influencing the relationship between High Performance Work Systems (HPWS) and work engagement. This relationship was proved to be positive in previous researches but the authors of the manuscript go further and consider an additional moderator which can affect this dependence. As former researches ignored the impact of negative social behavior at work, certainly a research gap exists, and the paper is an important contribution to this topic. The authors formulated two hypotheses on both indirect and direct association between HPWS and work engagement. The research is based on data from a survey conducted in Finland and is quantitative in nature. Several statistical methods were used to verify the hypotheses, namely Latent Class Cluster Analysis and Moderation Mediation Analysis. Using advanced methodological approach is a strength of the paper. Interesting results are factually presented and widely discussed, thus I consider this paper as a valuable contribution to evaluation of the importance of being bullied at work in regard with the effect of HPWS on work engagement. However it should be kept in mind that this study encounters certain limitations, particularly the imbalance of the sample according to gender.

Some detailed remarks:

  1. Table 2. As the title of the table explains, it includes the characteristics of the concepts. It is true for points 4 to 6, and to some extent 7. But Age, Gender and Supervisor position in my opinion are not ‘concepts’ or at least are different notions than HPWP, Organizational justice and Engagement which are derived from evaluations of statements of specific measurement scales. Moreover such features as Gender and Supervisor position are categorical/binary variables and correlations (I guess Pearson’s correlation was used?) between them as well as between them and other variables numeric in nature are a bit problematic. Moreover, the interpretation of the results in table 2 given by the authors does not concern Gender or Supervisor position but is concentrated on ‘concepts’, so please consider reducing the table 2 to ‘concepts’ or abandon correlations between binary variables as it is not a good choice for comparing them.
  2. Lines 225-227: there is an unexpected almost empty line – it should be corrected
  3. Lines 315-325. I appreciate the use of more sophisticated statistical technique (LCCA) than ‘classical’ ANOVA but as LCCA is not so well-known in health studies, additional information on this approach would be an advantage. I suggest including information that it differs substantially from ‘classical’ cluster analysis as it is a model-based approach and gives better premises to find a reasonable number of classes.
  4. Line 482: there is an accent mark (used in French language) over “a” in the word “and”
  5. Lines 564-566: The authors say “Also, we inspected method bias by comparing the fit of our theoretically expected factor model to a CFA including a common method factor prior to our analyses testing our research hypotheses.” but no result of this comparison is given. If such a procedure was performed it would be great to give a short comment of the outcome.
  6. The names of sections of the article are in capital letters except “Introduction”. It should be unified.
  7. Finally, a remark about citing (references). The journal uses a different way of giving references than this used in the manuscript, i.e. with consecutive numbers and not authors’ names and year of publication. According to IJERPH Instructions for Authors: “References must be numbered in order of appearance in the text (including table captions and figure legends) and listed individually at the end of the manuscript”.

Reviewer 2 Report

First of all, I would like to thank you for the opportunity to read your interesting paper entitled “High Performance Work Systems, Justice, and Engagement: Does Bullying Throw a Spanner in the Works?”  I think that you are tackling a timely and relevant topic, which deserves attention in the scholarly debate. This is an exciting and well-conceived study of important constructs, i.e., high performance work systems, bullying, justice, and engagement. The authors used the data of 434 employees of Finnish service.

Although the paper focuses on essential concepts and their relationship, a few concerns deserve attention. I list here in the spirit of offering some suggestions for improving the manuscript.

The introduction sounds confusing to me. The authors are unable to identify a major gap in the scientific knowledge they are going to fill in with their research. This prevents them from giving a “shape” to their research, which is presented in a confused and unattractive way in the current version of the manuscript. I recommend the authors carefully rewrite their introduction, trying to: 1) emphasize the gap in the scientific knowledge they are going to fill; 2) stress the relevance of their work and their distinctive contribution to the advancement of scientific literature; 3) clarify the research questions that are addressed in this research; 4) refine the scope of the article, avoiding to adopt a broad perspective which falls short in attracting the interest of the readership.

Please justify the need for the study in the introduction. The authors should focus more on addressing what we already know about the topic before bringing in a gap considering what the paper tries to fill in. This would make it clear to the reader why it is crucial to address the shortcomings in the literature.

It is not very clear why you chose these constructs and not others. I believe that your contribution would be more significant if you presented convincing arguments regarding why you chose these constructs and not others.

Much more is needed to justify how the predictor and moderator are related to the outcome variable. What new insight is your study offering to readers? We already know the relationship among the study variables in various domains, samples, and contexts. If you convince the reader of the necessity for your framework to expand our current knowledge, you would significantly enhance your contribution.

Please justify in detail how Affective Events Theory is best in grounding the study hypothesis. 

Please add the moderated mediation model for more clarity. 

I suggest reviewing past studies, identifying all the outcomes of HPWS, and listing them in a table with relevant literature review sources to understand this phenomenon better. You could introduce a pre-version of this table in the introduction in the form of a paragraph. With the help of this one, one can easily understand what we know and what we don’t know.

The literature review is not satisfactory; hence, more works seem necessary to bring it up to date. 

The research methods also present the main problem. The cross-sectional approach adopted by the authors cannot be considered consistent with the peculiar aims that inspired this article. Authors should carefully explain how a cross-sectional research design can be considered consistent to investigate the complex interplay between study variables A longitudinal research, at the moment, seems to be the most consistent research methodology with the research purposes undertaken by the authors.

Can you describe how participants were recruited? More information is needed on questionnaire design, and sample size selection. Participants and procedures should need to elaborate in more detail.

Please check the data normality, multicollinearity, and autocorrelations.

Please add the CFA for all the study variables, compare the measurement model with the alternate models, and then perform convergent and discriminant validity among variables.

Please justify why PROCESS macro is better than SEM.

The discussion, theoretical and managerial implications part is underdeveloped. In particular, the discussion needs to be more thorough and linked to your literature review.

The conclusion is a mere synthesis of the research findings. It is underdeveloped and it falls short in stressing and arguing the original contribution of this research. It should be completely revised, trying to emphasize the value of this research.

As several flaws can be retrieved throughout the manuscript, further proofreading of the paper is warmly recommended. For example, is it Finnish or Finland in the abstract?

I suggest that the authors incorporate recent and more context-related articles related to variables.

In general, I would like my recommendations to help the authors improve their work. I hope the authors will benefit from these suggestions and make the necessary amendments to strengthen the manuscript for later submission.

Round 2

Reviewer 2 Report

Dear Authors, I carefully re-evaluated your paper, finding it substantially improved with respect to the version. The revised version is much better organized and has higher scientific quality. Therefore, I recommended it for publication. Thank you